# The Fluidic Connectome in Brain Disease: Integrating Aquaporin-4 Polarity with Multisystem Pathways in Neurodegeneration

**DOI:** 10.3390/ijms262311536

**Published:** 2025-11-28

**Authors:** Felix-Mircea Brehar, Daniel Costea, Calin Petru Tataru, Mugurel Petrinel Rădoi, Alexandru Vlad Ciurea, Octavian Munteanu, Adrian Tulin

**Affiliations:** 1Puls Med Association, 051885 Bucharest, Romaniamugurradoi@innbn.com (M.P.R.); profciureaav@umfcd.ro (A.V.C.); octavianmunteanu@umfro.com (O.M.); 2Department of Neurosurgery, “Carol Davila” University of Medicine and Pharmacy, 050474 Bucharest, Romania; 3Department of Neurosurgery, “Victor Babes” University of Medicine and Pharmacy, 300041 Timisoara, Romania; 4Department of Ophthalmology, “Carol Davila” University of Medicine and Pharmacy, 050474 Bucharest, Romania; 5Central Military Emergency Hospital “Dr. Carol Davila”, 010825 Bucharest, Romania; 6Department of Vascular Neurosurgery, National Institute of Neurology and Neurovascular Diseases, 077160 Bucharest, Romania; 7Medical Section, Romanian Academy, 010071 Bucharest, Romania; 8Neurosurgery Department, Sanador Clinical Hospital, 010991 Bucharest, Romania; 9Department of Anatomy, “Carol Davila” University of Medicine and Pharmacy, 050474 Bucharest, Romania; adrian.tulin@umfcd.ro

**Keywords:** aquaporin-4 polarity, glymphatic system, neurodegeneration, neuroinflammation, cerebrospinal fluid biomarkers, extracellular vesicles, single-cell transcriptomics, multi-omics integration, precision medicine, artificial intelligence

## Abstract

The way in which Aquaporin-4 (AQP4) is localized on the astrocytes’ surface—i.e., with AQP4 channels predominantly located on the endfeet of astrocytes near the blood vessels—represents an important structural element for maintaining brain fluid homeostasis. In addition to this structural function, AQP4 polarity also facilitates glymphatic transport, the maintenance of the blood–brain barrier (BBB) functions, ion buffering, and neurotransmitter removal, and helps regulate neurovascular communications. The growing body of literature suggests that the loss of AQP4 polarity—a loss in the organization of AQP4 channels to the perivascular membrane—is associated with increased vascular, inflammatory, and metabolic disturbances in the context of many neurological diseases. As a result, this review attempts to synthesize both experimental and clinical studies to highlight that AQP4 depolarization often occurs in conjunction with early signs of neurodegeneration and neuroinflammation; however, we are aware that the loss of AQP4 polarity is only one factor in a complex pathophysiological environment. This review examines the molecular structure responsible for maintaining the polarity of AQP4—such as dystrophin–syntrophin complexes, orthogonal particle arrays, lipid microdomains, trafficking pathways, and transcriptional regulators—and describes how the vulnerability of these systems to various types of vascular stress, inflammatory signals, energy deficits, and mechanical injury can lead to a loss of AQP4 polarity. Furthermore, we will explore how a loss of AQP4 polarity can lead to the disruption of perivascular fluid movement, changes in blood–brain barrier morphology, enhanced neuroimmune activity, changes in ionic and metabolic balance, and disruptions in the global neural network synchronization. Importantly, we recognize that each of these disruptions will likely occur in concert with other disease-specific mechanisms. Alterations in AQP4 polarity have been observed in a variety of neurological disorders including Alzheimer’s disease, Parkinson’s disease, multiple sclerosis, traumatic brain injury, and glioma; however, we also observe that the same alterations in fluid regulation occur across all of these different diseases, but that no single upstream event accounts for the alteration in polarity. Ultimately, we will outline emerging therapeutic avenues to restore perivascular fluid transport, and will include molecular-based therapeutic agents designed to modify the anchoring of AQP4, methods designed to modulate the state of astrocytes, biomaterials-based drug delivery systems, and therapeutic methods that leverage dynamic modulation of the neurovascular interface. Future advances in multi-omic profiling, spatial proteomics, glymphatic imaging, and artificial intelligence will allow for earlier identification of AQP4 polarity disturbances and potentially allow for the development of more personalized treatment plans. Ultimately, by linking these concepts together, this review aims to frame AQP4 polarity as a modifiable aspect of the “fluidic connectome”, and highlight its importance in maintaining overall brain health across disease states.

## 1. Introduction: From Peripheral Symptom to Central Driver

Aquaporin-4 (AQP4) has historically been thought of as a “housekeeping” molecule that allows for water to be transported into or out of cells. It has been well studied as a result of its role in maintaining the proper fluid balance within cells (osmotic equilibrium) and reducing the swelling (edema) caused by excess fluids; however, it has been largely ignored in terms of its relationship to the various mechanisms that contribute to cognition and neurodegeneration, as well as how it contributes to the progression of disease. Historically, the large concentration of AQP4 located at the perivascular astrocytic endfeet was viewed as a simple structural feature, with no known physiological importance [1]. As a result, most of the research on brain pathology has been based upon a primarily neuron-centric model, with the focus being on misfolded proteins, failed synapses, lost neurons, and other clear indicators of degeneration. Changes in glial polarity have generally been seen as a secondary effect of disease, as opposed to a primary contributor to the disease process itself [2]. However, the tide is changing. Recent advances in single cell transcriptomics, high resolution imaging, and molecular genetics indicate that AQP4 is not simply a passive structural element but is an active player in whole-brain homeostasis [3]. There is mounting evidence that AQP4 mislocalization (commonly referred to as depolarization) is not just a random occurrence but an early, dynamic event that influences many physiological domains, including protein aggregation, vascular stability, and cognitive functions [1,4]. Furthermore, there is growing evidence that the degree of AQP4 depolarization in some disease models is correlated more with the disease trajectory than the final distribution of aggregated proteins, indicating that the organization of the astrocytic water channels is an upstream modulator of neurodegenerative processes [5]. Astrocytes have historically been viewed as inactive, non-participating “support” cells of the nervous system. However, recent studies have demonstrated that astrocytes are extremely dynamic and play a significant role in regulating the interstitial environment of the CNS, specifically regarding fluid transport, metabolism, and neurovascular communication [6].

Astrocytes are positioned throughout the CNS, and their perivascular endfeet represent specialized interfaces that allow them to continuously interact with the vasculature and surrounding neural tissue. Through these endfeet, astrocytes coordinate the movement of blood, cerebrospinal fluid (CSF), interstitial fluid (ISF), and nutrients, and facilitate the movement of ions and small molecules across the perivascular space [7]. AQP4 plays a critical role in facilitating water exchange across the perivascular space, thus facilitating the glymphatic system, which facilitates the removal of metabolic waste, misfolded proteins, and inflammatory mediators. The efficiency of the glymphatic system relies heavily upon the precise localization of AQP4. Misplacement of AQP4 results in disrupted convective flow, slowed clearance, and the promotion of the accumulation of pathologic aggregates [8].

The precise localization of AQP4 is maintained by a complex molecular scaffold, consisting of dystrophin–syntrophin complexes, agrin, and cytoskeletal elements that attach the channel to the perivascular membrane. AQP4 exists in two isoforms (M1 and M23) that exist in orthogonal arrays and regulate water permeability, and their distribution is regulated by transcriptional, post-translational, and epigenetic events [9,10]. Astrocytes can dynamically alter the localization of AQP4 in response to neuronal activity, vascular signaling, and circadian rhythms. For example, during periods of sleep, astrocytes increase their polarization to enhance glymphatic flow; whereas, during wakefulness, astrocytes favor ionic and neurotransmitter homeostasis. Therefore, the ability of astrocytes to dynamically regulate AQP4 localization increases the vulnerability of the system to disruptions. Vascular pathology (e.g., hypertension, ischemia) disrupts the endothelial signals and extracellular matrix composition that provide the necessary anchoring environment for the maintenance of AQP4 polarity [11]. Similarly, inflammation, cytokine exposure, age-related declines in mitochondrial functions, membrane lipid changes, and epigenetic drift impair the scaffolding of astrocytes and lead to the depolarization of AQP4 [12]. Loss of AQP4 polarity results in decoupling of the CSF and ISF compartments, thus slowing the clearance pathways and leading to the accumulation of amyloid-beta, tau, alpha-synuclein, and TDP-43 [13]. These pathologic aggregates then damage astrocytic endfeet and the vascular basement membrane, creating a self-reinforcing cycle of polarity loss. Altered fluid distribution results in disrupted osmotic gradients and hydrostatic balance, promoting localized edema and disrupting ionic homeostasis, which is essential for synaptic transmission. Additionally, depolarization disrupts the interaction between astrocytes and endothelial cells, thus weakening the BBB and allowing inflammatory mediators to infiltrate the CNS and exacerbate polarity loss [14]. The activated microglia release cytokines and reactive oxygen species that further promote the cascade of events leading to chronic inflammation in proximity to the vascular interface. These alterations do not stop at the level of individual astrocytes, but extend to neuronal networks. Astrocytes regulate extracellular potassium and glutamate levels, and loss of polarity prevents their ability to maintain excitatory balance. The subsequent depolarization and abnormal firing patterns and disrupted oscillatory states interfere with sharp wave ripples, gamma rhythms, and slow wave activity, all of which are necessary for memory consolidation and cognition [15,16]. Therefore, the network-level disturbances that occur as a result of the loss of polarity may continue long after the initial molecular pathology occurs and contribute to progressive functional decline. Of equal importance is the fact that AQP4 depolarization occurs relatively early in multiple disease models, often before measurable cognitive impairment or substantial protein accumulation. The restoration of polarity through genetic, pharmacological, or environmental means improves clearance, reduces aggregation, and mitigates behavioral deficits even if pathology has already developed [17,18,19]. New imaging technologies currently allow for the visualization of perivascular water flux, and potentially could identify polarity loss prior to overt clinical disease. Novel molecular tracers with specificity for AQP4 may be used to monitor channel distribution in vivo, and therefore allow for early staging and assessment of treatment efficacy [20].

Loss of AQP4 polarity has been documented in Alzheimer’s disease, Parkinson’s disease, amyotrophic lateral sclerosis (ALS), chronic traumatic encephalopathy (CTE), and cerebral amyloid angiopathy, among others [21,22,23]. While each of these diseases has distinct etiologies, they all demonstrate characteristics of glymphatic dysfunction, compromised blood–brain barrier integrity, neuroinflammation, and disrupted synaptic functions. Viewing these commonalities as the result of polarity collapse provides a unified conceptual framework to explain why vascular injury accelerates degeneration, why sleep disruption exacerbates disease, and why immune changes often occur prior to the development of protein aggregates [24].

This represents a significant shift in the way we think about neurodegenerative disease—from discrete cellular defects to interconnected failures across glial, vascular, immune, and network domains [25]. Recognizing AQP4 as a dynamic regulator of fluid transport, blood–brain barrier functions, and neuroimmune balance, rather than a passive architectural component, will fundamentally change our understanding of neurodegenerative disease. While loss of polarity is likely not the only factor in the development of pathology, its involvement in multiple physiological domains suggests that early detection and targeted restoration of polarity may offer therapeutic potential. Conceptualizing neurodegenerative disease as a systemic failure of fluid dynamics and cellular organization, as opposed to a fixed degenerative endpoint, opens up new avenues for intervention, monitoring, and the preservation of brain health [26]. As a result of these factors, the purpose of this article is to create a comprehensive theoretical structure through which we understand AQP4 polarity to be at the center of maintaining stability between vascular, immune, metabolic, and network stability, and to investigate how losing polarity leads to systemic susceptibility to neurodegenerative diseases. This review will synthesize the large body of evidence into a single theoretical structure that describes how losing polarity connects initial physiological alterations with eventual pathological states rather than function as a review of individual findings that are unrelated. Therefore, the primary question guiding this research is as follows: how far can AQP4 polarity loss be understood as a reversible upstream point that relates failures in fluid dynamics to both the onset and increase in neurodegenerative processes? As such, this review will describe the existing knowledge on the mechanisms of this process, define areas where there still exist unknown dependencies in the polarity network, and provide the foundational concepts necessary for developing new biomarkers and targeted therapies. In order to integrate the aims of this review, an organized narrative approach has been adopted. The relevant studies were identified in a search of the large scientific database (PubMed, Scopus, and Web of Science) through the combination of the keywords for “AQP4”, “polarity”, “glymphatic circulation”, “neurovascular unit”, and “neurodegeneration”. Peer-reviewed experimental, clinical, and imaging studies have been considered with the highest priority, which address mechanisms of polarity, disruptions in polarity, and/or the implications of the disruption of polarity on a system-wide level. Genetic, molecular, and computational data were also used to contribute to the conceptual synthesis of the work when such data supported the conceptual synthesis of the review. The exclusion criteria were based on the relevance to polarity-dependent physiology as opposed to rigid methodological criteria. In this way, diverse forms of evidence could be incorporated into a coherent theoretical framework. Although a formal systematic review was not the objective of the project, a clear, reproducible process for selecting and interpreting the evidence was developed in order to identify the most influential and mechanistically informative results. Thus, this semi-structured methodological approach allows the conclusions drawn herein to represent both the breadth of the existing literature and the broad conceptual relationships that unite what were previously observed as separate phenomena.

## 2. Molecular Blueprint of AQP4 Polarization: Anchoring, Regulation, and Dynamics

The polarized localization of AQP4 to the endfeet of astrocytes is not merely a result of the anatomical organization of tissue, it is the result of a highly organized molecular system consisting of cytoskeletal scaffolding, membrane specializations, interactions with the extracellular matrix, variations in isoform expression, trafficking pathways, and multi-layered regulating systems. The combination of all of these structures creates a special perivascular environment that allows for directional water flow between CSF and ISF and for the convective transport required for glymphatic circulation. Understanding the structural and regulating aspects that allow for the maintenance of polarity will be crucial for understanding how this organization can be disrupted by disease [27].

### 2.1. Anchoring Architecture: The Dystrophin–Syntrophin Scaffold and Perivascular Niche

Anchoring AQP4 requires the presence of the dystrophin-associated protein complex (DAPC). The DAPC anchors the astrocytic cytoskeleton to the vascular basement membrane. The C terminal PDZ binding site of AQP4 binds with α-syntrophin allowing AQP4 to become anchored in the DAPC. Dystrophin Dp71 acts as a bridge between the DAPC and the actin filaments. Agrin, laminin, and perlecan act as bridges between the DAPC and the ECM via β-dystroglycan [28]. All of these relationships are dynamic and respond to both mechanical and biochemical stimuli from the vasculature. Agrin derived from endothelium is important for creating the perivascular ECM niche where AQP4 is clustered. Aging, diabetes, or vascular stress reduce agrin and therefore weaken anchoring and lead to depolarization. Other scaffold components, including α-dystrobrevin, various syntrophin isoforms, spectrins, and ankyrins, contribute to the anchoring of the DAPC to the cytoskeleton and the recruitment of signaling partners and membrane curvature [29].

### 2.2. Isoform Diversity and Orthogonal Arrays: Structural Determinants of Polarized Function

The function of AQP4 is influenced by isoform composition. Alternative translational start sites produce two primary isoforms, M1 and M23, which vary in their ability to form orthogonal arrays of particles (OAPs): M23 easily forms large OAPs, while M1 forms small oligomers limiting lattice growth [1,30].

The ratio of M1/M23 influences OAP size and stability, and thus influences water permeability and membrane organization. The ratio of M1/M23 is controlled by multiple mechanisms, including upstream open reading frames, RNA binding proteins controlling transcript stability and localization, and post-translational modifications affecting the oligomerization rate. OAPs preferentially form in perivascular microdomains aligned with the microvilli of astrocytes, facilitating the directionality of water flow [31,32]. The organization of OAPs allows for the interaction with Kir4.1 potassium channels and facilitates ion/water homeostasis. Impairment of the balance of isoforms or the formation of OAPs will disperse AQP4 throughout the plasma membrane, reduce the directionality of water transport, and reduce glymphatic clearance [33].

### 2.3. Vesicular Trafficking and Membrane Domain Segregation

The exact delivery and retention of AQP4 at the endfeet require coordination of vesicle traffic. New synthesis of AQP4 is delivered from ER through the Golgi apparatus and is targeted to the perivascular membrane by Rab GTPase-regulated pathways, with Rab11 directing recycling endosomes to the astrocyte endfeet [34]. Kinesin and dynein motor proteins facilitate the transport along microtubules for longer distances, while actin filaments facilitate the final positioning of the membrane. The targeting of AQP4 depends on PDZ-containing sorting signals and, once in the membrane, AQP4 is confined to laterally segregated compartments. Lipid rafts rich in cholesterol and sphingolipids, caveolin, and flotillin assist the organization of these domains and constrain AQP4 diffusion [35]. Changes in lipid composition common to aging and metabolic disorders reduce the integrity of lipid rafts and increase the diffusibility of AQP4. Pathways for endocytosis also influence polarity: clathrin-dependent endocytosis removes incorrectly located AQP4, while the caveolae-dependent process controls the turnover of AQP4 at the endfeet. Repeated inflammation or oxidative stress will disturb these pathways and cause the widespread mislocation of AQP4 [36,37].

### 2.4. Multilayered Regulation: Transcriptional, Post-Translational, and Cytoskeletal Control

The polarity of AQP4 is determined by interrelated regulatory systems operating at the level of transcription, epigenetics, post-translational modification, and the cytoskeleton. Transcriptional regulation involves hypoxia-inducible factors, inflammatory transcription factors such as NF-κB, and circadian rhythm-related regulators such as BMAL1 and CLOCK [38]. Epigenetic regulation, including DNA methylation, histone modification, and the regulation of non-coding RNA affects the level and stability of transcripts. The post-translational modification of AQP4 modifies the behavior of channels: phosphorylation influences the organization and opening characteristics of OAPs; ubiquitination labels AQP4 for degradation if it is misplaced; S-nitrosylation modifies the interaction of AQP4 with the DAPC [39]. Components of the cytoskeleton integrate these signals. Astrocytic actin filaments serve as tracks for the directed movement of vesicles containing AQP4; microtubules are involved in the long distance transport; intermediate filaments such as GFAP contribute to the maintenance of structure of the endfeet [40]. Activity-dependent pathways involving rho GTPases and Ca^2+^ dependent pathways influence cytoskeletal remodeling in response to cues from neurons and vasculature. Chronic inflammation, oxidative stress, and chronic metabolic stress disrupt these mechanisms and make the system prone to depolarization [41,42].

### 2.5. Dynamic Remodeling: Spatial and Temporal Adaptation of AQP4 Microdomains

Astrocytic AQP4 microdomains are continuously being remodeled by neuronal activity, metabolic needs, vascular dynamics, and circadian rhythms. High resolution imaging studies demonstrate that the clusters of OAPs vary in size, density and location over time (minutes to days). Circadian rhythm play an important role: reduced noradrenergic activity and increased interstitial volume during sleep promote the clustering of AQP4 and enhance glymphatic clearance, while wakefulness promotes the ion buffering and reorganization of AQP4 according to requirements [43]. Chronic sleep disturbance disrupts these fluctuations and impairs clearance efficiency, and represents one of the earliest features of neurodegenerative diseases [44].

Rapid, Ca^2+^-dependent phosphorylation cascades induced by neuronal activity modify the anchoring interactions of AQP4, while mechanotransduction from vascular pulsatility aligns water flow with hemodynamic forces. Decreased mechanosensitive signaling due to vascular stiffening results in decreased perivascular stability [45]. Recent studies using spatial transcriptomics have demonstrated astrocytic heterogeneity with cells adjacent to arterioles expressing more genes involved in the anchoring of AQP4 than cells adjacent to venules, indicating local specialization in maintaining polarity [46].

### 2.6. Systems Integration: A Fragile Equilibrium of Interdependent Modules

Modules involved in the maintenance of polarity of AQP4—anchoring complexes, isoform ratio, trafficking pathways, compartmentalization of membranes, and cytoskeletal dynamics—form an extremely connected system. The disturbance of any module is likely to affect other modules: changes in the isoform ratio affect the architecture of OAPs; inflammatory phosphorylation of α-syntrophin will weaken the anchoring of the DAPC; changes in lipidome will impair the organization and stability of lipid rafts and channels [47].

Mathematical models describing the system as exhibiting threshold-dependent behavior demonstrate that the compensatory mechanisms will work until the density of anchoring or membrane integrity fall below a critical value, after which depolarization will accelerate rapidly [48]. Simulations of the systems integrating molecular interactions, membrane mechanics, and fluid dynamics confirm that there exists such a tipping point, demonstrating that slight disturbances can induce complete collapse of the system once structural buffers are exceeded [49].

Understanding AQP4 polarity as an emergent systems property highlights both its fragility and potential for recovery. Due to its proximity to critical thresholds, even minor improvements in anchoring, trafficking, and/or membrane stability will restore coordinated water flow and glymphatic functions [50]. Table 1 summarizes the most relevant structural and regulatory modules forming AQP4 polarity, highlighting the dependence of each module on the others. Identifying this integrated blueprint of AQP4 polarity forms the basis for developing therapeutic strategies for both the prevention of depolarization and the restoration of functional polarity once lost.

### 2.7. Conclusions of Section 2

AQP4 polarization emerges as an extremely precisely adjusted, integrated molecular blueprint which allows for the construction of scaffolding complexes, the composition of isoforms, vesicular trafficking, membrane organization, control and signaling, and dynamic modification. This attitude, however, implies fragility, for which a degree of adjustment in even one element will induce instabilities upon the whole network, and depolarization hence will ensue [30]. Understanding this blueprint is therefore essential to study how this component has been forfeited and what are the leverage points for therapy. That this machinery, this very molecular blueprint, would serve no other purpose in the long term but dynamically activate, does not only misplace the water channel, but would destroy a parent regulatory nexus upon which all clearance will depend, as will vascular communication, balance of immune phenomena, and cellular functions through which all communications must be conducted.

When combined, these molecular and organizational aspects demonstrate that AQP4 polarity is not merely an architectural preference, but rather a highly regulated and dynamic state dependent upon inputs from cytoskeletal, transcriptional, and membrane anchoring mechanisms. As all of these components work together to function as a cohesive system, disruptions within any one component have the potential to cause instability throughout the entire system. Therefore, it logically follows that understanding how this complex architecture is compromised in vivo would be the first step toward explaining why polarity loss occurs so frequently prior to the manifestation of early symptoms of neurological diseases. Consequently, the next section will discuss the stresses (i.e., vascular, mechanical, inflammatory, metabolic, and genetic) which disrupt the regulatory systems discussed previously and initiate the earliest stages of AQP4 depolarization.

## 3. Collapse Unleashed: Triggers and Early Events in AQP4 Depolarization

When AQP4 molecules are anchored in the astrocyte’s endfoot, they direct water flow through perivascular pathways for glymphatic transport. The polarization of AQP4 molecules is precise, but it is fragile. Stresses from various environmental sources, including physical, vascular, inflammatory, metabolic, genetic, and environmental, can disrupt the anchor sites, disorganize the cell membrane, reprogram gene expression, and disintegrate cytoskeletal networks. Many of these initial disruptions occur before the first signs of clinical dysfunction appear, and thus provide some clues as to why AQP4 depolarization can serve as an early indicator of increased disease vulnerability and a potential site of intervention to prevent disease progression [62,63,64].

### 3.1. Vascular and Mechanical Stressors: Hemodynamic Perturbation and Blood–Brain Barrier Breakdown

Under normal physiologic conditions, the dynamic shear generated by the pulsatile blood flow provides stabilizing forces for dystrophin/syntrophin anchoring, and OAP clustering at the perivascular region. However, if hemodynamic balance is disrupted either due to chronic hypotension or small vessel disease, the stiffness of the vasculature decreases the pulse pressure, the endothelium produces less agrin, and the β-dystroglycan–ECM interaction is weakened. As a result, the Dp71/α-syntrophin complex is weakened and AQP4 diffuses out of the endfoot and into the surrounding membrane, resulting in the disruption of unidirectional water flow [65]. Further disruption of the blood–brain barrier (BBB), through endothelial damage, can lead to rapid depolarization of AQP4. This occurs through the disruption of the basement membrane by endothelial cells allowing plasma proteins and thrombin to interact with astrocytes and stimulate intracellular cascades that degrade dystrophin/syntrophin complexes and reorganize the actin cytoskeleton [66]. Additionally, even short-lived disruptions to the BBB can cause significant mislocalization of AQP4 within hours indicating the close association between vascular integrity and maintaining polarity [67].

Traumatic injury has also been shown to cause AQP4 mislocalization through multiple pathways. Traumatic injury can cause the shearing of astrocyte endfeet disrupting cytoskeletal connections, dispersing AQP4 clusters, and causing microhemorrhage and the subsequent activation of reactive gliosis. All of these injuries create large areas where AQP4 has lost its proper orientation [68]. Altogether, these data demonstrate that vascular stability and mechanical force represent major regulatory pathways for AQP4 localization, while other peripheral influences affect AQP4 through these central pathways [69].

### 3.2. Inflammatory and Immune Mechanisms: Cytokine Cascades and Glial Signaling

Cytokines released by activated microglia can rapidly disrupt AQP4 polarity. Upon binding to astrocytic receptors, cytokines such as IL-1β, TNF-α, and IFNγ can initiate the activation of NF-κB dependent transcriptional programs that upregulate AQP4; however, cytokines also induce the downregulation of anchoring components and the degradation of agrin and laminin by matrix metalloproteases, which separates the number of channels from their spatial distribution [70].

Activated astrocytes also undergo transitions to GFAP and vimentin enriched phenotypes that alter the morphology of the terminal processes of the astrocyte. Complementary proteins, such as C1q and C3, accumulate in the perivascular space of astrocytes and induce astrocytes to transition to a GFAP/vimentin-rich phenotype that disrupts the geometry of the terminal processes of the astrocytes. The inhibition of complement signaling prevents the disruption of polarity caused by complement signaling [71]. Leukocytes entering through a damaged or compromised BBB can deliver cytokines and reactive oxygen species to astrocytes, suppressing alpha-syntrophin and Dp71 and disrupting the lipid rafts necessary to maintain AQP4 clustering. Chronic exposure to cytokines also disrupts astrocytic calcium signaling, destabilizes actin filaments, and impairs vesicular transport, all of which disrupts the perivascular microdomains and slows glymphatic transport [72,73].

### 3.3. Genetic and Epigenetic Determinants: Predisposition and Molecular Vulnerability

Genetic variability affects the degree of stability of the AQP4 polarity network. Mutations in AQP4 alter the relative proportion of M1 and M23 isoforms, modify the density of OAPs, or disrupt the C-terminal PDZ binding sites necessary for anchoring to α-syntrophin. Certain haplotypes have been identified to be associated with decreased levels of M23 mRNA and/or altered phosphorylation of AQP4, leading to decreased lattice stability and increasing the risk of depolarization after exposure to stress [74,75].

Mutations in SNTA1 or DAG1 can disrupt PDZ-mediated binding or ECM attachment to AQP4, resulting in a widespread and non-perivascular distribution of AQP4 and decreasing perivascular water flow. Although each mutation alone may not significantly impact basal physiology, each can lower the threshold for depolarization by environmental or physiological stressors. Epigenetic mechanisms can also confer additional vulnerability to disease. Hypermethylation of the promoter of AQP4 can decrease transcriptional responsiveness to stress, and histone modification of genes involved in anchoring and trafficking can vary depending on the metabolic and inflammatory state of the cell. Non-coding RNA (such as miR-29 and miR-130) can regulate both the AQP4 and SNATA1 mRNAs and influence the response to metabolic and systemic stressors [76,77].

### 3.4. Aging and Metabolic Signaling: Gradual Erosion of the Polarity Network

Aging results in gradual and cumulative effects on the AQP4 polarity network. Decreased mitochondrial efficiency limits the energy available for transporting vesicles and remodeling cytoskeletons. Oxidative stress can also modify dystrophin–syntrophin and cytoskeletal proteins that weaken anchoring and allow the lateral diffusion of channels [78,79].

Age-related alterations in metabolic signaling can also compromise AQP4 polarity. Decreased levels of insulin/IGF-1 can decrease the transcriptional responsiveness of AQP4. Alterations in cholesterol and sphingolipid composition can also disrupt the lipid raft stability and increase the movement of channels. Decreased circadian rhythm amplitude with age decreases clustering during sleep and reduces glymphatic transport. Each of these factors can independently compromise the AQP4 polarity network and collectively create a narrower margin of safety that can lead to rapid failure when challenged by secondary insults [80,81].

### 3.5. Environmental and Systemic Insults: Trauma, Toxins, and Peripheral Inflammation

Environmental insults can precipitate or exacerbate the loss of polarity. Cumulative shear forces caused by repetitive head injury can induce microhemorrhages, ECM remodeling, and chronic expansion of depolarization [82,83]. Exposure to airborne particulates can activate inflammatory signaling in both the systemic circulation and endothelial cells, alter the composition of the basement membrane, and disrupt the BBB functions. Heavy metals can disrupt calcium homeostasis and actin filament dynamics in astrocytes, disrupting the cytoskeletal structures that support the membrane domains [84].

### 3.6. Convergence and Early Dynamics: Synergy Among Triggers

The early dynamics of the loss of polarity arise from the nonlinear interactions between vascular injury, inflammation, metabolic dysfunction, and genetic predisposition. Vascular stiffness and focal leaks in the BBB enable the entry of leukocytes into the CNS; inflammation can degrade ECM components and anchoring complexes, thereby creating mechanical instability; metabolic disorders can disrupt the lipid domains and trafficking pathways; and genetic background can determine the thresholds at which these failures will occur. The combination of these factors creates positive feedback loops that can greatly accelerate depolarization once initiated [85]. The mislocalization of AQP4 often occurs before the measurable accumulation of amyloid-beta or cognitive impairment. Modern advanced imaging techniques can detect subtle polarity disruptions following vascular injury or systemic inflammation in individuals who are at risk. Mathematical modeling demonstrates that maintaining polarity exhibits threshold properties. Once the density of anchoring or the stability of the membrane fall below a certain level, depolarization can proceed rapidly regardless of the magnitude of prior insults. These predictions are consistent with the observation that restoring polarity is most effective when initiated before the system crosses this tipping point [86]. Figure 1 illustrates the combined molecular and physiological forces that destroy the polarity network long before overt pathology develops and emphasizes the synergistic and mutually reinforcing relationships between them.

### 3.7. Conclusions of Section 3

The polarity of AQP4 seems to be dependent on a fine balance of forces which can be easily upset, hemodynamically, inflammatorily, genetically, metabolically, and by environmental influences. These factors are synergistic and induce loops of self-reinforcing which intensify the associated depolarization and prepathology to devolve and proceed to be pathology in a detectable form. The insight and targeting of the mechanisms conducive to the loss of polarity may allow a chance of an intervention being performed whilst the history is still as such in a wholly reversible condition [87].

These initial destabilizers collectively show that AQP4 depolarization is not simply an isolated molecular event, but represents the first “tipping-point” for a larger system(s) dysfunction. The tightly-regulated integration of perivascular water flow, astrocyte communication, and vascular elasticity, once disrupted through loss of polarity, begins to fall apart. This transition will serve as the physiological link between the aforementioned stressor(s) which initiated this process and the subsequent large-scale hydraulic dysfunction(s). In order to determine how a localized aberration in channel position ultimately results in the widespread compromise of brain fluid transport and clearance, we examine the effects of AQP4 dysfunction on glymphatic organization and pressure dynamics in the next section.

## 4. Glymphatic System Disintegration: The Hydraulic Consequences of AQP4 Failure

The loss of AQP4 polarity leads to a reorganizing of brain fluidics: the flow of directional convection becomes fragmentary, pressure transmission decreases, solute processing is slowed, steepening of ion and pH microenvironments occur, remodeling of perivascular destruct is noted, and the dynamics of the network becomes unstable [88]. The change appears to involve a type of systemic phase change from regulated convection to diffusion-driven transport, converting local molecule defect into a global dysfunctional state [89].

### 4.1. Disruption of Perivascular Fluid Architecture and Pressure Dynamics

Under physiologic conditions, the phenomenon of arterial pulsatility connects with the perivascular spaces, driving CSF into periarterial spaces and affecting its exchange with ISF via the AQP4-rich endfoot processes of the astrocytes with a low resistance pathway [90,91]. The amplitude of flow and reach depend on vascular compliance and effective water permeability. However, during depolarization, permeability becomes patchy by the insertion of high resistance segments, defeating the continuity of hydraulic transmission. The pulsatile energy is wasted and dissipated before it becomes effective, the pressure gradients become modified, CSF influx from the blood vessels and perivenous return decrease [92].

Computer models indicate that a slight decrease in perivascular conductance can cause the disproportionate loss of velocity and clearance, corresponding to a nonlinear relationship of flow resistance on the basis of branching geometries. Pulse transmission is dampened and phase-shifted, with corresponding effects on mechanotransduction in astrocytes and pericytes, with disordered nitric oxide signaling and disordered vascular tone. Thus, defects in signaling for fluidic and vascular impulses develop together from the same structural defect [93,94].

The relationship of AQP4 polarity to CSF movement creates a specific axis of vulnerability in the brain’s fluidic anatomy. AQP4 localized to the perivascular endfeet of astrocytes facilitates the conversion of the directionally defined inflow of CSF near arteries into a similar directional inflow into the surrounding parenchyma; in doing so, it allows for the propagation of arterial pulsations through the low resistance pathway connecting the periarterial space to the parenchyma [95]. This organization provides a stable hydraulic link between the inflowing CSF and the outflowing (both venous and lymphatic) CSF/ISF. The depolarization of AQP4 causes a loss of polarity leading to the redistribution of AQP4 from the perivascular endfeet of astrocytes across the rest of the astrocyte membrane; in doing so, the areas of reduced permeability lead to fragmentation of the CSF–ISF exchange pathways [96]. As a result, the inflow becomes disorganized and variable in terms of the magnitude and direction of convective flow to the various parts of the parenchyma; in addition, flow will occur primarily through diffusion rather than convection [97].

Tracer experiments have shown that loss of AQP4 polarity results in a non-uniform geometry of CSF penetration into the parenchyma, resulting in irregular wavefronts of the tracer and decreased penetration of the tracer into deeper structures of the brain. The irregular geometry of CSF penetration results in an increased time for solutes to be cleared from the parenchyma, an increased incidence of stagnant regions, and an increased susceptibility to fluctuations in blood vessel tone [98]. The CSF–ISF pressure gradient, normally maintained by the pulsatile nature of the vasculature and facilitated by the polarized water flux of the astrocytes, becomes less stable due to the loss of AQP4 polarity. As a result, the perivascular space will become enlarged, or collapsed in a segmental manner, due to the lack of coordinated water transport across the astrocyte membrane. The decreased hydraulic coupling will decrease glymphatic inflow, reduce the consistency of perivenous return, and render lymphatic clearance through the meninges less efficient [99].

While the loss of hydraulic coupling between the CSF and ISF will result in slowed fluid movement, it will also affect the mechanotransduction of astrocytes, pericytes, and vascular smooth muscle cells, all of which require coherent CSF–ISF dynamics to maintain normal levels of nitric oxide signaling, regulation of vascular tone, and maintenance of metabolic support to the parenchyma. Therefore, the depolarization of AQP4 will modify not only the flow of CSF, but the functional relationships that exist between the flow of CSF and the signaling and regulatory mechanisms of the vasculature and parenchyma [100]. Understanding the role of AQP4 polarity as a structural determinant of CSF–ISF exchange, therefore, will help clarify how molecular alterations at the level of the astrocyte endfoot can alter fluid dynamics throughout the glymphatic–venous network [62]. This schematic (Figure 2) illustrates how AQP4 polarity organizes directed CSF inflow and maintains coherent CSF–ISF exchange. Loss of polarity disrupts hydraulic coupling, fragmenting glymphatic flow and destabilizing vascular–glial signaling.

Although AQP4 orientation is primarily responsible for directing CSF flow to the periarterial and parenchymal regions, it is clear, however, that AQP4 organization is a different mechanism than that governing ISF removal. The three mechanisms involved in regulating ISF efflux are as follows: (1) polarity-directed endfoot permeability gradients, (2) anisotropic water conductance through the astrocyte membrane, and (3) the presence of a perivenous pressure gradient to drive ISF outflow, which is independent of the CSF inflow pathways [101]. Therefore, disrupting AQP4 polarity will disrupt ISF clearance through changes in hydraulic resistance within the brain tissue, loss of endfoot-driven convective transport, and disruption of coherence of the venous outflow system. In order to understand how nanoscale disruptions at the astrocyte endfoot result in macroscopic disruptions of glymphatic–venous transport, a detailed understanding of these polarity-dependent mechanisms is required [102].

### 4.2. Failure of Convective Solute Transport and Self-Propagating Accumulation

Convection is essential in the clearance of any and all of the macromolecules possessing low diffusion coefficients in a dense neuropil. Since the convective Peclet numbers fall with depolarization of the pulsatile arterial blood flow, the motion affected by transport becomes diffusion-limited, and the local concentrations of amyloid-β, tau, input, TDP-43, etc., accumulates with progressive increase with the rousting of nucleation and fibrillization. These aggregates then obstruct the perivascular waterways, causing increased resistance and pushing further down the flow [103]. They also magnify the activation of microglia and the release of proteases which degrade the basement membranes and anchoring proteins leading to the architectural collapse: feed forward in which the inability of the products of metabolism to be removed exacerbates the barriers to removal of products of metabolism. Beyond the proteinopathy, however, is an accumulation of lactate, protons, and reactive oxygen species acidifying the local pH, altering the functions of channels and receptors, and oxidizing neutral and cytoskeletal products which maintain stabilization of the endfeet. The chronic presence of cytokines results in glial reactivity. The interstitium becoming from the buffering functions which maintained homeostasis to a milieu which is pro-pathogenic, which enhances the degenerative cascades [104].

### 4.3. Temporal Collapse: Disintegration of Circadian Clearance Rhythms

Due to the reduction in adrenergic tone during sleep, there is an increase in interstitial volume and the oscillatory control of AQP4, an increase in perivascular clustering and consequent convective fluxes, and an increase in the removal of products of metabolism. The oscillatory function is diminished because of the depolarization during the period of night. The nocturnal peaks of removal of products of metabolism are diminished, but are still present in the interval of natural awakening, accumulating across the cycles [105]. The abnormal rhythms of the solutes are deranging to neuromodulatory systems of, for example, the adenosine signaling, causing the fragmentation of sleep and a decrease in synaptic homeostasis. The altered adrenergic and transcriptional programs feedback to maintain depolarization, which results in a bidirectional loop between the altered sleep and the hydraulic failure. This loop is consistent with the observations that disruptions of sleep precede and worsen cognitive decline [106,107].

### 4.4. Structural Remodeling of Perivascular Spaces: From Fluid Stasis to Architectural Failure

Fluid stasis alters the anatomy of the perivascular niche. The diminished shear, diminished endothelial nitric oxide, favors deposition of the matrix in the extra-cellular space and a thickening and stiffening of basement membranes with diminished compliance caused by propagated pulses. Pericytes are contractile and decrease perivascular volume and mechanical conditions for endfeet, which also shifts AQP4. The immune dynamics also change to a slower turnover of cytokines, and DAMPs give a prolonged activation of NY macrophages and astrocytes [108]. Chronic low-level inflammation promotes the matrix of metalloproteinases which downregulate astrocytic endothelial signaling and gives rise to disorders of the blood–brain barrier, additional immune infiltration, and ECM remodeling. This is as far as heteroscedastics go, exaggerated closer to the penetrating arterioles where the most consequential flow deficits occur, and gives rise, again plausibly, to alterations in the topology of aggregate depositions seen in disease, including the alteration of excluded solute gradients, ionic content, and altered neurovascular signaling [109,110]. Convection also aids the maintenance of the ionic homeostasis of the extracellular compartment and with distribution of metabolites and signaling molecules. With depolarization, the microdomains of increased potassium and glutamate concentrations occur leading to a delay of repolarization and hyperexcitability permitting NMDA dependent excitotoxicity. The pH is also rendered less stable leading to altered enzyme kinetics and receptor sensitivity. These changes lead to a breakdown in neurovascular coupling. The potassium and nitrous oxide gradients around vessels normally regulate the smooth muscle tone and local perfusion. Their distortion leads normally to a lessened fidelity of blood flow regulations to neuronal activity limiting the access of oxygen and substrates to captured circuits. Diminished pulsatility and diminished convection are added limits the feedback loop in the dysregulation of vascularity—failure of fluid clearance [111]. Disruption of fluids lead to network hydraulic collapse linked to non-normally functioning neural circuits. The network of the extracellular milieu is for regulatory interface circuit dynamics. With the loss of a convective buffer, the fractional volume in the extracellular milieu, the ionic homeostasis and metabolite distributions are wider ranging. The coherence of oscillatory activity is lost: gamma synchrony is reduced, decreasing attentional and working memory power; slow wave activity is fragmented, leading to impairments in memory consolidation [112].

The “fluidic connectome”—the network of perivascular and interstitial routes which modulate neuronal computation by maintaining the extracellular milieu—is disrupted with AQP4 depolarization. Multiscale models coupling the transport of fluids with network electrophysiology predict a transition from stable oscillatory regimes to irregular or hypoactive states with only modest changes in convective capacity. Regional deficiencies of flow give rise to spatially localized network deficits, providing the mechanistic link between hydraulic failure and gradated deterioration of cognition [97].

### 4.5. Translational Implications: Biomarkers, Therapeutics, and Diagnostic Frontiers

Hydraulic collapse has immediate implications for therapy and diagnosis. The manner in which drugs are distributed in the brain parenchyma is partly sufficiently dependent on perivascular routes; impaired convection leads to reduced penetration and altered pharmacokinetics of both intrathecal and some systemic agents. Restoration of polarity might enhance drug delivery by reestablishing convective routes [113].

Similarly the interpretation of biomarkers is dependent upon the profile of clearance dynamics. The soluble indicators of pathology—amyloid-β, tau fragments, and inflammatory mediators—have concentration–time profiles determined by glymphatic transport. Taking account of polarity-dependent clearance will improve staging and aid early diagnosis. Advances in imaging, such as dynamic contrast MRI and molecular tracers, are now available, which visualize the influx of CSF and the clearance of ISF in vivo, showing flow fragmentation consistent with depolarization [114]. The application of these modalities to computational models of hydraulic failure may make possible the recognition of preclinical hydraulic failure and the introduction of polarity-restoring interventions [115]. Based upon this, Table 2 attempts to synthesize how AQP4 depolarization reshapes the therapeutic and diagnostic landscape.

### 4.6. Conclusions of Section 4

AQP4 depolarization produces a rearrangement of brain hydraulic structure. Water transport in its directional modes becomes fragmented; the transmission of pressure energy is decreased; the clearance of macromolecules becomes diffusion limited; ionic and metabolic microdomains become destabilized; peripheral solid architecture is remodeled; coherence of networks collapses. The outcome of this is a transformation from fluidic homeostasis to fluid-driven pathology. Placing neurodegeneration within this context raises polarity from being a detail of membranes to a determinant of systems. Those strategies which restore perivascular AQP4 and reestablish convective exchange may remodulate vascular, immune, and network functions allowing for a tractable means of modifying morbid trajectories [124].

Once the system switches from organized or coordinated convective fluid transport to a disorganized or fragmented, diffusion-limited pattern of fluid movement, AQP4’s depolarizing effect on the cell membrane changes from being a localized hydraulic defect to a mechanism for causing widespread physiological instability. Once the mechanisms for fluid exchange, pressure transmission, and metabolic buffering become non-coherent in their function, they will begin to cause an increase in disturbances that move outward from perivascular spaces into vascular, immune, and neural domains. The transformation between impaired glymphatic flow and multisystem pathology is thus provided by this breakdown in hydraulic integrity. The next section addresses how the hydraulic defects caused by polarity collapse lead to the widespread physiological deficits described above by examining the vascular regulatory processes, blood–brain barrier functions, immune responses, metabolic pathways, and integrative networks affected by the loss of hydraulic coherency.

## 5. Systemic Fallout: How AQP4 Collapse Rewires Brain Pathophysiology

Collapse of the polarized state of AQP4 has a profound effect on the organization of the brain’s physiology far beyond the disruption of the local water balance. Once the perivascular architecture breaks down, a systemic, hierarchical cascade of changes occurs in vascular signaling, blood–brain barrier integrity, immune regulation, metabolic equilibrium, and network dynamics.

Therefore, AQP4 depolarization is not simply an extension of the failure of the glymphatic system, but rather a structural reorganization of multiple physiological hierarchies linking various pathological states through the same mechanisms of fluid, blood–brain barrier, and network dysfunction [91].

### 5.1. Neurovascular and Blood–Brain Barrier Breakdown: From Local Instability to Systemic Dysfunction

Astrocytes, endothelium, pericytes, neurons, and the extracellular matrix form a unit called the neurovascular unit (NVU). Each component relies on perivascular AQP4 to coordinate astrocytic mechanotransduction with vascular pulsatility. Loss of polarity will weaken calcium signaling, reduce nitric oxide and prostaglandin release, and ultimately cause the NVU to lose the ability to regulate vascular flow, and thus lead to localized areas of mild tissue hypoxia despite no apparent arterial disease [125]. Tight junctions between endothelial cells are compromised and, as such, the paracellular pathway becomes less selective, allowing plasma proteins to enter the parenchyma where they can activate astrocytes and microglia. The endothelium begins to upregulate vesicular transport and leukocyte adhesion molecules in response to the increased permeability. Detachment of pericytes from the basement membrane reduces the space surrounding the vessels, impairs mechanosensing due to altered shear forces, and disrupts glucose and lipid delivery to the parenchymal cells. As such, the blood–brain barrier function is now a conduit to pathology, and energy metabolism, myelination, and synaptic function are severely impaired [126,127]. As the degree of permeability continues to increase, it allows for additional signals from peripheral immune cells to propagate low-grade inflammation within the CNS that will continue to degrade junctional integrity, forming a positive feedback loop. Thus, breakdown of the NVU, remodeling of the BCB, and hemodynamic stresses each exacerbate the loss of polarity [128].

Studies using AQP4 knockouts have expanded this concept. While the loss of AQP4 clearly affects the glymphatic flow, solute clearance, and the calcium signaling of astrocytes, the disease-specific effects of AQP4 loss are still incompletely understood. For example, AQP4 knockout models exhibit accelerated amyloid-beta and tau accumulation and increased microglial activation in Alzheimer’s models, increased dopaminergic vulnerability in α-synucleinopathy models, decreased potassium buffering and metabolic support in demyelinating and motor neuron disorder models, although the causal relationships are still unclear [129,130,131].

Similarly, AQP4 genetic studies in humans show similar results: restriction fragment length polymorphism (RFLP) and regulatory variants of AQP4 have been shown to affect susceptibility to edema, cognitive resilience during aging, and sleep dependent glymphatic efficiency. Variants have also been associated with an increased risk or severity of Alzheimer’s disease, NMOSD, and TBI. Presumably, variants alter water permeability, anchoring density, and/or stress responses; however, the penetrance and functional impacts of the variants are still unknown [132]. Collectively, both knockout and genetic studies suggest that variations in the expression or localization of AQP4 result in the loss of NVU stability, blood–brain barrier competence, and the chemical environment in which neurodegeneration occurs [133].

### 5.2. Immune Activation, Ionic Collapse, and Network Instability

The continuous removal of cytokines, complement signaling, and DAMPs by polarity-dependent convective flow will prevent the accumulation of pro-inflammatory and pro-degenerative substances in the parenchyma. However, when the removal of these substances is disrupted, the accumulation of these substances will induce microglial activation, release of IL-1β, TNF-α, ROS, and proteases, which will break down the synapses, the extracellular matrix, and ultimately, neuronal integrity. Astrocytes will also develop neurotoxic programs, including the release of complement signaling, decrease in glutamate uptake, and reduction in metabolic support for neurons [134].

The cytokine environment will suppress the transcription of proteins responsible for anchoring AQP4 to the cell surface, while increasing the production of matrix metalloproteinases, resulting in the degradation of the extracellular matrix, while ROS will destabilize cytoskeletal and membrane components of AQP4 clusters, thereby continuing the inflammatory process [135,136].

At the same time, the ionic and metabolic homeostasis will deteriorate. Decreased convection will lead to an accumulation of potassium ions and delay repolarization and hyperexcitability of neural networks. Increased glutamate will drive NMDA receptor-mediated calcium overloading, excitotoxic cascades, and pH instability, while exhausted astrocytes will fail to buffer calcium ions and regulate blood flow [137]. Energy demands will increase due to elevated levels of ATP, while impaired mitochondrial functions, due to oxidative stress, will worsen the metabolic deficiencies [138]. At the network level, the convergence of these molecular failures will result in fragmentation of gamma synchronization, deterioration of slow wave activity, and loss of coherent communication between distant parts of the brain. Mechanisms of plasticity (LTP/LTD) will fail to operate under conditions of ionic and metabolic instability, and cognitive deficits will arise prior to substantial neuronal loss [139]. These CNS disruptions will also interact with peripheral immune and neuroendocrine systems. The disruption of drainage through the meningeal lymphatics will lead to altered antigen presentation and weakened immune surveillance, whereas the accumulation of CNS cytokines in the systemic circulation will reinforce chronic peripheral inflammation, which will feed back into the brain through endothelial and vagus nerve pathways [140,141,142].

The disrupted hypothalamic and brainstem detection of CSF solute gradients will lead to the disruption of rhythmic processes of appetite, arousal, autonomic tone, blood pressure, and metabolic functions—frequently observed features of many neurodegenerative disorders, but insufficiently explained by localized neurodegeneration [143].

### 5.3. Convergence Toward Disease: Shared Logic Across Disorders

Across various scales, AQP4 depolarization produces common pathophysiological logic. Loss of NVU function and BCB dysfunction create an opportunity for immune invasion; the subsequent inflammation will dismantle the structures that maintain AQP4 anchoring; ionic imbalance and excitotoxic stress will modify network dynamics; and feedback loops through peripheral tissues will extend the scope of dysregulation to include regions outside the CNS [144]. This common structure explains the presence of AQP4 depolarization, glymphatic dysfunction, neuroinflammation, and vascular instability in Alzheimer’s disease, Parkinson’s disease, multiple sclerosis, traumatic brain injury, and numerous other disorders [145]. Looking at the progressive degradation of brain physiology through this integrated perspective, the differences between these disorders appear to be largely a matter of varying degrees of a common degenerative sequence initiated or exacerbated by AQP4 polarity collapse. Therapeutic interventions targeting AQP4 polarity may provide a means to influence disease trajectory across diagnostic categories and serve as a conceptual anchor for multi-system therapeutic strategies [146]. Figure 3 attempts to illustrate this sequence: perivascular depolarization disrupts communication between the NVU and vascular pulsatility → chronic inflammation and metabolic deficiency accumulate → ionic and synaptic instability occur → network coherence is lost → peripheral immune and neuroendocrine systems are entrained. Thus, the hierarchical architecture of the above sequence establishes AQP4 polarity as a key determinant of systemic neurological decline, and possibly as a potential target for reversal of neurological decline.

### 5.4. Conclusions of Section 5

The loss of polarity of AQP4 is afflicted with stupendous effect from the local perivascular membranes to that of the consideration of the much greater architecture of the function of the brain. It is disassembly of the coordinated functionalizing of the NVU, it remodels the blood–brain barrier, it stimulates chronic inflammatory processes, it destroys ionic and metabolic equilibrium, and it causes loss of synchrony within the genome of the nervous system. It brings about a loss of immune functions, it wipes out wired neuroendocrine feedback, and their resultant coupling between loss of central function and systemic loss of control. The formed results, we imply, should show a remapping of the lost control of polarity, close to that of systems level defect, and suggest that restoration of this would naturally involve the simultaneous readjustment of vascular, immune, metabolic, and neurological factors [133].

In combination, this disarray in systems-level functions illustrates that AQP4 depolarization profoundly alters the overall functioning of the brain; it is not limited to the interface between the blood vessels and the brain. Once, in turn, vascular signaling, immunological equilibrium, metabolic homeostasis, and neuroendocrine feedbacks all deviate from their usual coordinated levels of function, the organism will be in a condition where its compensatory systems are no longer able to maintain the level of disruption to the organism’s systems. This widespread alteration of the organisms’ systems creates an environment in which the numerous diseased pathways will have ample opportunity to develop, combine, and mutually reinforce each other. In the next portion of this review, we examine how the above described large-scale physiological disruptions lead from early impairment to cascading disease processes, and explain why the loss of polarity so frequently represents the transition from reversible dysfunction to chronic pathology.

## 6. From Collapse to Cascade: Pathological Consequences and Disease Amplification

The loss of AQP4 polarity does not constitute an isolated event; it serves as the starting point for a self-reinforcing, nonlinear pathologic cascade of events leading to the failure of clearance, the onset of inflammation and vascular dysfunction, metabolic stress, and, ultimately, the collapse of the neural network [147].

### 6.1. From Clearance Failure to System-Wide Amplification

Impairment of the glymphatic system leads to the accumulation of amyloid-β, tau, α-synuclein, and TDP-43 outside of the cellular environment, thereby altering their kinetic behavior towards nucleation and subsequent fibril formation. Aggregates form within the perivascular space and impede the flow through the space. As aggregates accumulate, they stimulate glial receptor activation resulting in sustained inflammatory signaling [148]. Released proteases generated during this inflammatory response cause the degradation of laminin and agrin, destabilize the perivascular scaffold, and further reduce clearance capabilities generating a self-reinforcing cycle of protein deposition and structural breakdown. A restoration of AQP4 polarity has been shown to decrease the amount of aggregate formed within the perivascular space, indicating the role of flow dynamics in disease progression [149].

Stagnated solute loads and inflammation propagated throughout the vasculature lead to the development of vascular pathology. Oxidative and inflammatory responses to cytokines and hypoxia damage endothelial cells. Aberrant fluid dynamics also alter ionic gradients that are necessary for vascular reactivity, leading to an inability to regulate blood pressure, thus creating a state of hypoperfusion that generates oxygen glucose deprivation and mitochondrial failure. Mitochondrial failure and oxidative damage generate further oxidative damage and metabolic failure. Hypoxia-driven remodeling of the endothelium increases vascular permeability, allowing leukocytes to enter the CNS where they produce cytokines that support the inflammatory response [150]. Furthermore, hypoxia causes the apoptosis of oligodendrocytes disrupting the structure and function of myelin, thus affecting the integrity and function of white matter. Astrocytes become metabolically stressed and begin to rely on glycolysis for energy production, thus decreasing their ability to recycle neurotransmitters and buffer ions [151]. Accumulation of glutamate and potassium in the CNS can damage axons and synapses, and mitochondrial failure disrupts the transport of materials along axons, making white matter vulnerable to damage across all “gray matter” diseases [152].

At the network level, these failures converge to create an unstable network. The loss of gamma synchrony creates a lack of coordination between different parts of the network. Slow wave activity collapses creating a loss of the rhythmic coordination that was previously present. Long range communication becomes unreliable due to the energy dependent plasticity mechanisms failing due to excessive calcium and insufficient ATP and the deterioration of dendritic structure under oxidative stress. Importantly, these failures occur prior to the widespread loss of neurons and indicate that cognitive decline occurs due to the disequilibrium of fluids and ions rather than solely due to neuron death [153].

### 6.2. Disease-Specific Patterns and Nonlinear Dynamics

While polarity loss generates a common cascade, the disease context affects how the cascade is expressed. For example, in Alzheimer’s disease, impaired clearance of amyloid-β promotes tau propagation and vascular stiffening; in Parkinson’s disease, impaired glymphatic clearance fosters α-synuclein accumulation and dopaminergic vulnerability; in multiple sclerosis, perivascular inflammation and metabolic insufficiency worsen demyelination; after traumatic brain injury, mechanical disruption of endfeet creates prolonged polarity loss, edema, and cognitive dysfunction; in gliomas, altered perivascular dynamics transform the tumor microenvironment affect the blood–brain barrier functions [154].

Computational modeling demonstrates that, despite these differences, the commonality among the diseases lies in the nonlinear nature of the disease process. Specifically, once perivascular AQP4 density drops below a certain threshold, convective clearance fails, solute accumulates, and ionic regulation loses stability. Therefore, small changes in membrane conductivity or ion gradient concentrations rapidly lead to the sudden transition of network behavior from ordered, oscillatory behavior to irregular or hypoactive behavior [155]. These threshold effects can explain why many diseases exhibit an initial period of apparent dormancy followed by a rapid progression to severe symptoms, and why restoring polarity is most effective when performed before the critical transition points have been reached [156].

### 6.3. Conclusions of Section 6

Depolarization of AQP4 begins a multidimensional cascade of events in which clearance failure promotes aggregation, inflammation degrades structural scaffolding, vascular insufficiency induces metabolic collapse, and network coherence disintegrates. At some point, across many diseases, this represents the tipping point beyond which reversible dysfunction becomes irreversible degeneration. Viewing depolarized AQP4 as a malleable point provides new perspectives on neurological diseases as disorders of fluidic failure across systems. Thus, restoring polarity should be viewed not simply as a fortunate side effect of therapy, but as a strategically positioned therapeutic anchor with the capability to influence disease trajectory [157].

Thus, a therapeutic strategy aimed at this point of convergence will focus on restoring perivascular anchoring, maintaining trafficking, and restoring convective flow. By doing so, it may restore clearance, stabilize vascular signaling, and break the cycles of inflammation and metabolic failure that promote degeneration. The therapeutic strategy outlined in the next section is predicated on this rationale.

## 7. Therapeutic Frontiers: Restoring Polarity and Rebuilding Glymphatic Function

Recognition of AQP4 polarity as an adjustable factor in maintaining normal conditions of the brain has changed how therapeutic options have been approached. An increasing amount of evidence suggests that depolarization is the earliest point at which vascular, inflammatory, metabolic, and network disruptions start to mutually enforce one another. Approaches designed to restore polarity—whether by repairing structure, reprograming transcription, redirecting cell types, or using bioengineering to deliver materials—seek to restore directional water flow, normalize clearance, and interfere with disease processes prior to their becoming self-sustaining [158].

### 7.1. Rebuilding the Polarity Machinery: Molecular and Genetic Strategies

Repairing the dystrophin-associated complex (DAC) which anchors AQP4 to perivascular membranes is the basis for structural repair. Damaging dystrophin, α-syntrophin, β-dystroglycan, or related molecules disrupts endfoot anchoring and is a major pathway leading to depolarization. Stabilizing AQP4–α-syntrophin interactions with small molecules or preventing destructive phosphorylation increases channel retention and the speed of recovery of localized perivascular AQP4 [159,160]. Gene therapy vectors expressing functional forms of the DAC components delivered to astrocytes restore clustered AQP4 and improve the clearance of solutes in experimental settings. Modulating the cytoskeleton is an additional approach to affect the ability of AQP4 to move and to maintain perivascular location. Cytoskeletal organization greatly affects the ability of AQP4 to move, and it is vulnerable to damage from inflammatory and mechanical stress. Agents that stabilize filamentous actin or inhibit proteins involved in depolymerizing actin filaments, such as cofilin, support the maintenance of the DAC’s function to maintain polarity and endfoot architecture [161].

Gene editing technologies expand these approaches. Platforms based on CRISPR/Cas can edit deleterious mutations of dystrophin, syntrophins, or regulatory elements; synthetic transcription factors can selectively induce the expression of anchoring proteins in reactive astrocytes; and tools that modify the epigenome can make promoters accessible to support the sustained expression of programs that support polarity without changing the genome [162,163,164,165,166]. Together, these therapies provide precise correction of structural and regulatory vulnerabilities responsible for polarity failure.

### 7.2. Reprogramming Astrocytic States: Transcriptional, Epigenetic, and RNA-Based Modulation

Although the expression of the DAC contributes to structural polarity, it is controlled by the transcriptional and epigenetic states of the astrocytes. Neuroinflammation typically reduces the expression of AQP4 and the anchoring gene expression, and can alter the ratio of AQP4 isoforms. The use of transcription factors, such as NF-κB, STAT3, and CREB, can restore AQP4 expression, normalize the M1/M23 ratio, and increase the expression of scaffold proteins [167]. RNA-based therapies are complementary to these approaches. Antisense oligonucleotides and siRNAs can modify the splicing of AQP4 to favor polarity-supportive AQP4 isoforms. Synthetic mRNAs can increase the number of perivascular channels during periods when there is increased susceptibility. These tools allow for the state-dependent modulation of gene expression in astrocytes, and therefore allow for early, very specific intervention [168,169,170].

### 7.3. Targeted Delivery and Microenvironment Engineering: Nanomedicine and Biomaterials

Any therapies that will be able to correct polarity must be able to access the gliovascular interface—a major barrier to delivery. Nanomedicine offers several ways to address this issue by allowing targeted, condition-responsive delivery. Nanoparticles engineered to bind to endothelial or astrocytic ligands are taken across the blood–brain barrier via transcytosis and accumulate in perivascular compartments where they can release their cargo in response to local signals, such as pH, oxidative stress, or cytokine concentrations. This allows for the delivery of small molecules, RNA, or gene editing components to the site of action with high specificity and reduced systemic exposure [170]. Exosomes and engineered extracellular vesicles provide an alternative biocompatible delivery method by utilizing the natural mechanisms of crossing the BBB and minimizing the elicitation of an immune response. Engineered exosomes can be modified to bind to dystrophin or syntrophin, thereby enhancing the repair of anchoring structures, and can also be loaded with anti-inflammatory or trophic factors that modify the microenvironment in a way that favors the restoration of polarity [171]. Biomaterials provide opportunities for regeneration. Bioactive scaffolds containing astrocytic progenitor cells can populate damaged regions and reconstitute polarity-competent networks. Advances in 3D printing and microfluidic organoid systems now enable the rapid assessment of scaffold compositions that optimize anchoring integrity and perivascular organization [172].

### 7.4. Modulating the Inflammatory–Vascular Axis: Neuroimmune and Bioelectronic Therapies

Since inflammation and vascular dysfunction both initiate and perpetuate depolarization, modulating the inflammatory–vascular axis globally provides a second approach to therapy. Broad-spectrum inhibitors of NF-κB or JAK/STAT signaling decrease the activation of microglia and create a reparative environment that supports polarity, such as methods to reprogram microglia and astrocytes to promote anchoring stability and reduce matrix degradation [173].

Peripherally-activated inflammation can also affect polarity of the CNS through cytokine spillover and endothelial activation. The peripheral administration of TNF-α and IL-1β antagonists, metabolic anti-inflammatory regimens, and targeted vascular therapies can reduce inflammatory spillover across the BBB and indirectly protect glymphatic functions [174,175].

Non-pharmacologic modulation of the inflammatory–vascular axis is provided by bioelectronic methods. Vagal nerve stimulation decreases peripheral and CNS inflammation and promotes endfoot stability. Focused ultrasound can temporarily modulate perivascular water movement and BBB permeability to facilitate drug delivery and activate mechanotransduction pathways in astrocytes [176]. Chemogenetic and optogenetic modulation of astrocytes and vasculature further provide a means to precisely regulate vascular tone and fluid dynamics. Systems that monitor glymphatic flux in real time and adaptively stimulate to modulate flow represent an emerging field [177].

### 7.5. Precision Polarity Medicine: AI, Multi-Omics, and Clinical Translation

Precision medicine approaches are required due to the complexity of regulating polarity. Single-cell transcriptomic, spatial proteomic, lipidomic, and metabolomic multi-omics profiling is identifying new polarity-regulatory molecules and guiding combinatorial strategies that target multiple layers of regulation simultaneously [178].

AI is accelerating the translation of the findings to the clinic. Machine learning algorithms integrate molecular, imaging, and physiological data to predict which patients would benefit from therapies targeting polarity, to stratify disease subtypes based on glymphatic impairments, and to predict treatment responses [179]. AI-assisted drug discovery is identifying compounds that modulate polarity pathways, while digital twins of individual patients simulate the effects of interventions on perivascular flow and network coherence before clinical trials [180]. Computational fluid dynamic models further predict how interventions shape hydraulic behavior over all scales [93].

Because polarity collapse demonstrates threshold behavior, the timing of the treatments is critical. Several biomarkers of early polarity disruption—CSF/ISF tracer kinetics, perivascular diffusion metrics, and soluble AQP4 fragments—are being investigated to detect polarity disruption before the establishment of irreversible feedback loops [181,182]. Combinatorial therapies that combine structural stabilizers with immunomodulators, or gene editing approaches with biomaterials, are intended to correct all of the failure points simultaneously. Initial clinical studies are beginning to apply these concepts, including nanoparticle-based AQP4 modulators, focused-ultrasound protocols to enhance glymphatic flow, and systemic immunotherapies to target inflammation disrupting polarity [183].

### 7.6. Conclusions of Section 7

AQP4 polarity has developed as a tractable therapeutic axis that may be used to reshape disease progression. Until recently, polarity was considered a static structural feature of astrocytes. However, current knowledge indicates that polarity is a dynamic determinant of fluid transport, vascular signaling, immune tone, metabolic stability, and network functions. The diverse array of therapeutic approaches—from molecular engineering and reprogramming of astrocytes to nanomedicine, biomaterials, neuromodulation, and AI-guided precision therapy—converge to the objective of reestablishing directional water flow and glymphatic efficiency. Reestablishment of polarity has the potential to reverse the clearance deficits and vascular instabilities that contribute to disease, as well as to recalculate the systems physiology that sustains neuronal resilience [184].

While this review combines molecular, physiological, and systems level data to support AQP4 polarity as an essential regulatory mechanism of brain homeostasis, there are a number of limitations that should be identified.

Firstly, the majority of the information presently understood about AQP4 polarity failure was derived from rodent models and, therefore, can be subject to significant interspecies variability in terms of vascular geometry, astrocyte diversity, sleep patterns, and perivascular dynamics. This results in the inability to make direct translations from experimental findings on polarity failure, glymphatic flow, and disease progression in rodents to humans.

Secondly, the available data are highly segmented by scale. While the molecular data regarding dystrophin–syntrophin interaction, isoform regulation, and trafficking pathways are robust, the relationship between these mechanisms and macroscopic changes in hydraulic functions, vascular integrity, or network coherence is rarely directly demonstrated. Still in their infancy, high resolution imaging techniques are being developed, which would allow researchers to investigate the early loss of polarity, changes in perivascular resistance, or alterations in microdomain organization in humans.

Thirdly, the model presented in this review integrates a large number of different biochemical pathways, including inflammatory signaling, vascular remodeling, metabolic failure, and network instability; however, the hierarchical nature of these interactions and how they relate to each other in a temporal sense (i.e., cause and effect) remains poorly defined. Many of the associations reported in the literature are correlative; thus, while causality may be inferred based upon the timing of events in animal models or through indirect biomarkers, it cannot be established with confidence using longitudinal human data.

Fourthly, while depolarization is presented here as an early event in many neurological diseases, due to its convergence on common molecular mechanisms, this conceptually unifying framework remains purely theoretical. There are numerous disease-specific factors that influence the occurrence and impact of polarity loss, including genetic variation, comorbidity of vascular risk, immune profile, and environmental exposure(s), all of which have not yet been fully characterized.

Lastly, while emerging therapies, including gene editing, nanotechnology, biomaterial scaffolds, and bioelectronics, hold promise for restoring polarity in complex human disease states, most of these therapies are still in preclinical or early-stage translational development. Thus, the ability of these therapies to restore polarity in the complex environment of human disease and the safety of sustained modulation of fluid dynamic pathways will require thorough and long-term testing.

Overall, these limitations emphasize the need for multidisciplinary and multiscale research, the continued improvement of non-invasive imaging of perivascular behavior, and the determination of the causal relationships between polarity and pathology in humans, as well as the design and execution of early phase clinical trials. By recognizing these boundaries, the current synthesis can be framed as a working model which will serve to provide guidance for future study rather than a definitive resolution of the underlying mechanistic processes.

## 8. Conclusions and Future Perspectives: Toward a New Era of Fluidic Neurobiology

The asymmetric distribution of AQP4 at astrocytic endfeet is a curiosity, but more than that, it is an organizing principle of the brain. By virtue of its regular and precise localization, AQP4 affects water flux, drives the glymphatic flow, maintains ionic gradients, and couples vascular pulsatility and metabolic demand. Its failure initiates a cascade that transforms the fluidic environment of the brain from a well-ordered convective structure into a stagnant, diffusion-dominated entity. This cascade of change propagates outward, resulting in abnormal protein aggregation, chronic neuroinflammation, vascular decline, metabolic collapse, network disintegration, and cognitive decline—many of the common and devastating features seen in most of the major neurological diseases.

To recast polarity collapse in light of the unrevealing mechanism also invites a recasting of our body of knowledge on neuropathology. Rather than thinking of Alzheimer’s disease, Parkinson’s disease, multiple sclerosis, cranio-encephalic trauma, or glioma as fundamentally distinct, they ought to be regarded as a number of different and distinct manifestations of a common systems failure: the disintegration of polarity-determined fluid regulatory mechanisms. This suggests that the importance of abnormal protein folding, inflammatory mediated influences, and genetic risks are not irrelevant but may be viewed in a wider mechanistic framework, in the sense that fluidic dynamics and extracellular signals form important upstream determinants of the evolution of the pathology. Hence, we may assign to the extracellular space and its hydraulic properties foremost positions in pathophysiology and therapy.

The burgeoning discipline of fluidic neurobiology aims to map out, model, and manipulate this hidden dimension of brain functions. Much like what the connectome revolutionized our understanding of neural circuitry, the idea of a “fluidic connectome”—a unified interconnected perivascular and interstitial system operating by virtue of perivascular AQP4 polarity—promises to revolutionize our understanding of clearance, signaling, and homeostasis. Newer imaging modalities, such as dynamic contrast-enhanced MRI or light-sheet microscopy, are just beginning to visualize glymphatic flow in unprecedented detail, while the spatial omics approaches are painting newer insights into the molecular architecture of perivascular precincts and their cellular constituents. To integrate these datasets with computational models will afford predictive maps of fluid dynamics and the risk of illness and/or therapeutic responses. Therapeutic avenues are already leading to new directions reflecting this paradigm. Molecules exerting therapeutic effects work to enhance the stability of anchoring complexes as well as modulating isoform ratios in the tissue. Genome editing and epigenome editing procedures work to repair or fine-tune the machineries of polarity. Nanotechnology and biomaterials are being used for targeted delivery and structuring of perivascular niches. Neuromodulation strategies enhance vascular pulsatility as well as astrocytic signaling, which indirectly augments polarity. Increased integration of intervention sets is already being seen to augment greater end organ effects that can repair both the structure, molecular quantity, and systemic nature of the failed polarity simultaneously.

The future will be dominated by precision polarity medicine: a combination of molecular profiling, imaging biomarkers, as well as predictive modeling to assess risk, to guide therapy, as well as to monitor for response. Machine learning algorithms may eventually discover insights that could provide early signatures to indicate polarity disturbance, permitting presymptomatic intervention. Closed-loop therapeutic models may be introduced permitting real-time adjustment of interventional strategies based upon continuous tracking of glymphatic flow as well as perivascular homeodynamics. Such precise modalities must serve not only to interrupt the disease process but to effect prevention against disease in the first place. The systemic relevance of polarity extends well beyond the brain. The glymphatic flow modulates the interaction of immune, endocrine, and metabolic homeostasis. Its perturbation yields chronic inflammation, dysregulation of hormonal flow, and systemic metabolic imbalance. The restoration of polarity may therefore effect the recalibration in physiology, remaindering body physiology, creating new connections of neurology, immunology, and endocrinology in a new opportunity for interdisciplinary interventions.

But profound questions remain. What is the earliest molecular event which occasions the collapse of polarity? In what ways do vascular aging, state of metabolism, circadian rhythms, and other systemic factors play their part? Where are the bifurcations which separate reversible loss of polarity from the auto-accelerative descent into decline? How do fluid dynamics modify neural plasticity and cognition at the level of the system? And are interventions possible not only for repairing polarity but amplifying it beyond the original efficiency of its operation? For the solutions to these questions, there must be the deepest interfusion of collaborations between departments of molecular biology, bioengineering, physics, computational modelling, and clinical neuroscience—a sort of interdisciplinary synthetic directionality which will correspond to the very systemic paradigm of which polarity is a part. The progressively clear lesson that advances in this field are teaching us that the health of the brain is strictly reciprocal to, one might say, the flow dynamics and flow gradients which flow through it. Polarity is becoming clear, not as some peripheral dynamic, but as a major organizing principle of which instability would signal a transition from homeostasis to pathology, and restoration of polarity the manner in which that change from decline to recuperation would affect itself.

If the atrophy of AQP4 polarity is the moment in time between hydraulic order in the brain and its unwinding, its restoration may be the initiation of a new age, one characterized by the conscious manipulation of fluid dynamics for promotion, or at least inflation, of neural resilience. The near future then of liquid neurobiology should be such that not merely reparation will be thought of but a deeper understanding of the brain itself, an organ whose states of functioning and non-functioning are inscribed not in terms of synapses and circuits but the hidden currents of fluid which operate between them.

## Figures and Tables

**Figure 1 ijms-26-11536-f001:**
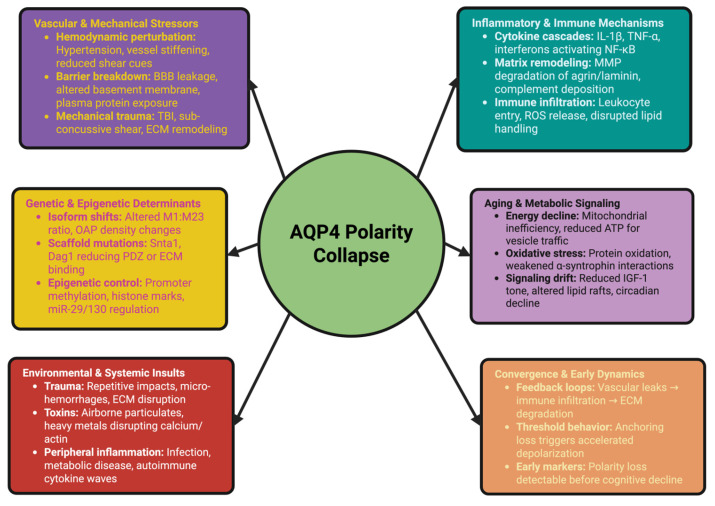
Triggers and early events driving AQP4 polarity collapse. Polarized AQP4 channels at astrocytic endfeet are essential for directing water flux along perivascular pathways and maintaining glymphatic circulation. This polarity is vulnerable to diverse stressors that act through convergent molecular routes long before clinical symptoms appear.

**Figure 2 ijms-26-11536-f002:**
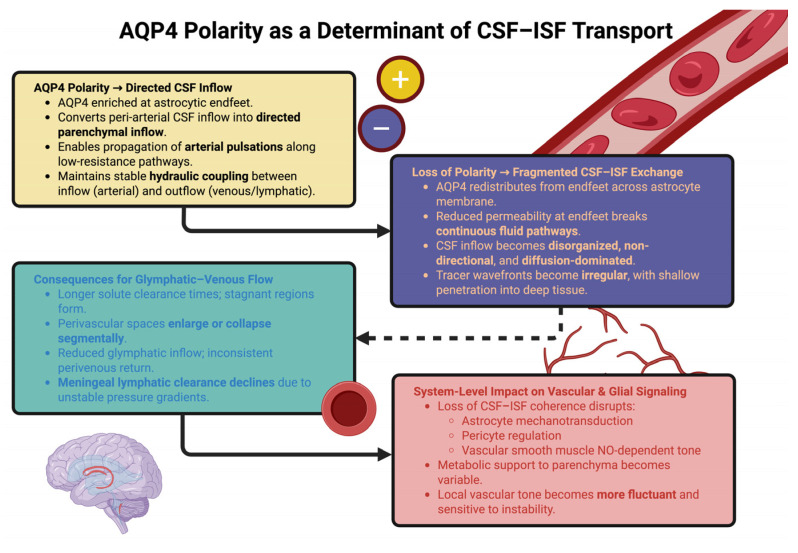
Polarized AQP4 at astrocytic endfeet channels periarterial CSF into structured parenchymal inflow, supporting pulsation-driven transport and stable pressure gradients.

**Figure 3 ijms-26-11536-f003:**
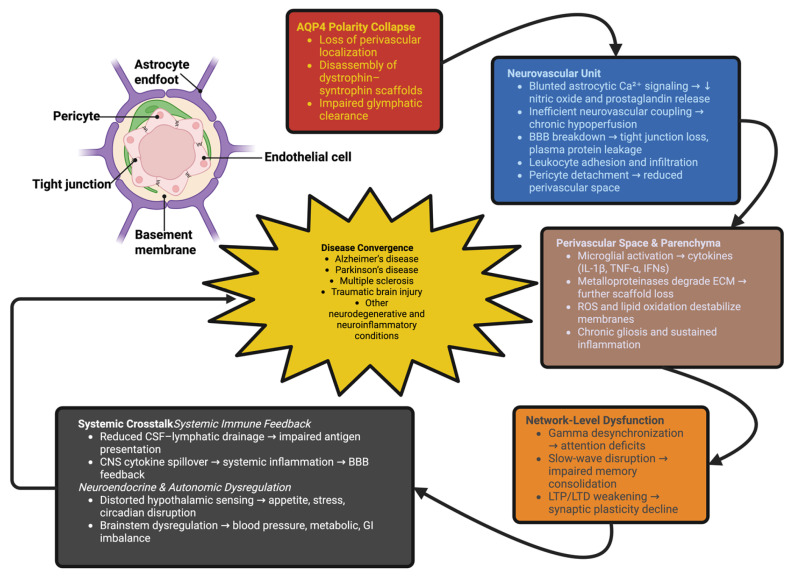
Cellular cascade and systemic consequences of AQP4 polarity collapse. Loss of perivascular AQP4 localization disrupts water homeostasis and glymphatic flow, initiating a cascade that propagates from the neurovascular interface to systemic physiology. (Neurovascular unit) Impaired astrocytic Ca^2+^ signaling weakens neurovascular coupling and compromises blood–brain barrier integrity, leading to plasma protein infiltration, leukocyte adhesion, and pericyte detachment. (Perivascular space) Microglial activation, cytokine release, extracellular matrix degradation, and oxidative stress amplify scaffold loss and sustain neuroinflammation. (Network level) Ionic imbalance, excitotoxicity, and mitochondrial stress destabilize neuronal signaling, disrupt oscillatory synchrony, and impair synaptic plasticity. (Systemic crosstalk) Impaired CSF–lymphatic drainage blunts peripheral immune responses, while cytokine spillover, hypothalamic dysfunction, and autonomic imbalance extend pathology beyond the CNS. (Disease convergence) These interconnected processes form a shared pathogenic framework linking vascular, immune, metabolic, and neural disruption in Alzheimer’s disease, Parkinson’s disease, multiple sclerosis, traumatic injury, and related disorders.

**Table 1 ijms-26-11536-t001:** Molecular architecture and regulatory logic of AQP4 polarization. The table seeks to provide a concise yet integrative overview of the molecular systems that establish and sustain AQP4 polarity at the astrocytic endfeet—a fundamental determinant of glymphatic water transport. It outlines the key modules involved, from anchoring scaffolds and isoform dynamics to vesicular trafficking, membrane microdomains, transcriptional and cytoskeletal regulation, mechanotransduction, and sleep-state modulation. Each component is presented alongside its core mechanism, representative molecular players, failure modes linked to disease, potential biomarkers, and emerging therapeutic levers. Together, these elements illustrate how polarity emerges as an emergent property of an interdependent network—one that is highly adaptable yet fragile, and whose disruption underlies diverse neurological disorders.

Regulatory Domain	Fundamental Role	Principal Components	Characteristic Disruptions	Indicative Measurements	Intervention Strategies	References
Perivascular Anchoring Complex	Secures AQP4 at astrocytic endfeet through ECM–DAPC linkage.	α-Syntrophin, Dp71, β-Dystroglycan, Agrin	Mislocalization; reduced perivascular density; impaired CSF–ISF coupling.	Endfoot AQP4/parenchymal AQP4 ratio; ECM marker profiling.	Scaffold-reinforcing peptides; agrin-based stabilization.	[51,52]
Isoform Configuration & OAP Formation	Determines lattice assembly and water permeability efficiency.	AQP4-M1/M23 isoforms; phosphorylation regulators	Loss of OAPs; diffuse membrane distribution; slowed clearance.	OAP structural assays; intrathecal tracer kinetics.	Isoform ratio modulators; OAP-stabilizing compounds.	[1,53]
Trafficking & Endfoot Targeting	Directs AQP4 vesicles to perivascular domains.	Rab11, Microtubules, Kinesin/Dynein motors	Endfoot delivery failure; ectopic accumulation near synapses.	Vesicle-targeting markers; Rab11 localization indices.	Vesicular routing enhancers; PDZ signal modifiers.	[54]
Membrane Microdomain Organization	Maintains localized AQP4 clusters within lipid-ordered zones.	Caveolin-1, Sphingolipids, Kir4.1	Increased lateral mobility; disrupted K^+^–water coupling.	Membrane order imaging; Kir4.1/AQP4 colocalization.	Lipid microdomain regulators; Kir4.1 co-support.	[55]
Transcriptional & Epigenetic Regulation	Controls Aqp4 transcription, rhythmicity, and inflammatory responsiveness.	HIF-1α, NF-κB, BMAL1, microRNAs	Aberrant expression; circadian flattening; edema susceptibility.	Transcript levels; circulating miRNA signatures.	Epigenetic modifiers; circadian-aligned dosing.	[56,57]
Cytoskeletal Integration	Supports structural stability and vesicle docking at endfeet.	GFAP, RhoA, CaMKII	Reactive gliosis; disrupted anchoring geometry; impaired trafficking.	CSF GFAP; actin remodeling assays.	Gliosis-modulating agents; cytoskeletal stabilizers.	[58]
Vascular & Mechanical Signaling	Aligns polarity with pulsatility, endothelial cues, and flow dynamics.	Pericytes, Piezo channels, eNOS	Polarity loss with vascular stiffening; reduced hydraulic coupling.	Perivascular MRI; vascular compliance metrics.	Vascular-compliance therapies; mechanosensitive stimuli.	[59]
Sleep–Wake Modulation	Enhances AQP4 clustering and clearance during low-NE sleep states.	β-Adrenergic receptors, cAMP	Reduced nocturnal clustering; impaired diurnal clearance.	Sleep-locked tracer clearance; EEG-linked flow metrics.	Slow-wave enhancement; chronotherapeutic timing.	[60,61]
Systems Level Integration	Maintains stability across interacting polarity modules.	Cross-module interactions	Threshold-dependent collapse; nonlinear transition to global depolarization.	Composite polarity indices; multi-parametric modeling.	Early multi-target intervention; staged restoration paradigms.	[15]

**Table 2 ijms-26-11536-t002:** Systemic Consequences of AQP4 Depolarization on Brain Fluidics and Network Homeostasis. This table summarizes how the loss of AQP4 polarity transforms brain physiology from the molecular to the systems level. It highlights the cascading disruptions that arise across key domains—from perivascular fluid dynamics and convective clearance to circadian rhythms, ionic microdomains, and large-scale network behavior.

Functional Axis	Physiological Contribution	Consequences of Polarity Loss	Indicative Measurements	Intervention Angles	References
Hydrodynamic Coupling	Converts arterial pulsatility into directed CSF transit along perivascular channels, supporting fluid exchange with parenchyma.	Fragmented flow pathways; reduced hydraulic reach; inefficient mechanotransduction by astrocytes and perivascular cells.	Attenuated influx waveforms on dynamic MRI; altered cerebrovascular reactivity profiles; NO-related metabolic signatures.	Reinforcement of perivascular ECM; stabilization of dystrophin-associated anchoring complexes; vascular compliance optimization.	[93,116]
Bulk Clearance and Proteostasis	Removes macromolecules through convection-dominant transport, maintaining extracellular compositional stability.	Collapse into diffusion-dominant kinetics; retention of misfolded proteins; ECM softening and secondary inflammatory remodeling.	CSF protein spectra (Aβ, tau, α-syn); neuroinflammatory PET markers; lactate and metabolite accumulation.	Modulators of AQP4 clustering; proteostasis-enhancing compounds; oxidative-stress attenuation.	[117,118]
Rhythmic Fluid Regulation	Aligns glymphatic activity with sleep–wake cycles, enhancing nocturnal solute turnover.	Blunted circadian variation; impaired adenosine dynamics; persistent adrenergic suppression of perivascular channel clustering.	Flattened diurnal solute oscillation curves; sleep–EEG architecture deviations; metabolic neuromodulator assays.	Circadian-phase–specific treatment; modulation of arousal circuits; stabilization of sleep-dependent clearance.	[119]
Perivascular Structural Homeostasis	Preserves basement membrane composition, vascular pliability, and controlled entry of immune mediators.	ECM thickening; pericyte dysregulation; low-grade vascular inflammation; increased paracellular permeability.	MMP activity assays; PDGFRβ release; GFAP/S100B elevation; MRI permeability shifts.	ECM remodeling strategies; pericyte-state modulators; endothelial–astrocyte communication repair.	[29,120]
Ion and Neurovascular Microdomain Coordination	Maintains spatiotemporal K^+^/glutamate control and couples neural activity to vascular adjustments.	Accumulation of excitatory ions; pH microdomain instability; impaired K^+^–NO–vascular feedback loops.	Extracellular ion and pH mapping; delayed hemodynamic response signals; vasoreactivity quantification.	Targeted ionic-buffering approaches; NO-pathway reinforcement; pH-stabilizing interventions.	[121]
Network-Level Integration	Stabilizes the extracellular milieu needed for synchronous oscillations and long-range neural communication.	Volatile extracellular volume; dampened gamma coherence; disrupted slow-wave organization; compromised network controllability.	EEG coherence metrics; functional connectivity MRI; computational control network indices.	Closed-loop neuromodulation; metabolic–fluidic hybrid therapies; modulation of astrocytic signaling programs.	[122]
Clinical Translation Interface	Ensures predictable solute kinetics relevant for biomarker analysis, therapeutic distribution, and disease staging.	Distorted biomarker clearance curves; uneven therapeutic dispersion; reduced diagnostic sensitivity of fluid-based measures.	Glymphatic flow indices; tracer washout kinetics; solute-distribution modeling.	Polarity-restoring molecular therapies; improved drug-routing techniques; AI-informed mechanistic modeling.	[123]

## Data Availability

No new data were created or analyzed in this study. Data sharing is not applicable to this article.

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
