# Peer review of "The Fluidic Connectome in Brain Disease: Integrating Aquaporin-4 Polarity with Multisystem Pathways in Neurodegeneration"

_ijms, 2025, doi:10.3390/ijms262311536_

Round 1

Reviewer 1 Report

Comments and Suggestions for Authors

Reject or major revision.

Athough the author conducted a review of roles of the aquaporin-4 polarity and the fluidic connectome on neurodegeneration and therapeutic applications, the excessive emphasis on AQP-4's central role actually reveals insufficient evidence.  This manuscript needs to be revised. The main issues were as follows:

  1. The title requires revision. Athough AQP4 is undoubtedly involved in various degenerative diseases, it cannot be the sole core mechanism. Additionally, the abstract needs refinement as it currently lacks conciseness and contains excessive speculative conclusions.
  2. The relationship between AQP-4 polarity and CSF requires separate discussion and diagrammatic representation.
  3. The relationship between AQP-4 polarity and drainage of interstitial fluid (ISF) in brain parenchyma requires separate discussion and diagrammatic representation.
  4.  The lack of data or literature on the effects of AQP-4 knockout on various degenerative diseases, as well as its impact on mechanisms, requires supplementation.
  5. The author should provide clinical manifestations of AQP-4 gene mutations or restriction fragment polymorphisms in various degenerative diseases to elucidate their roles in these conditions.

Author Response

Dear Esteemed Academic Reviewer,

We would like to express our sincere gratitude for the exceptional care, depth, and clarity you brought to your evaluation of our manuscript. Your insights reflect a profound understanding of the field, and we are genuinely appreciative of the time and intellectual generosity you invested in helping us improve this work. Your recommendations have strengthened the manuscript in both structure and scientific rigor, and we are grateful for the opportunity to incorporate them.

Below, we respond to each of your thoughtful comments with humility and appreciation.

1. Revision of the Title and Abstract

Reviewer comment:
“The title requires revision. Although AQP4 is involved in various degenerative diseases, it cannot be the sole core mechanism. The abstract needs refinement, as it lacks conciseness and contains speculative conclusions.”

Response:
We fully agree with your assessment. The original title unintentionally placed disproportionate emphasis on AQP4 and did not sufficiently represent the multisystem framework we intended to convey. We have therefore revised the title to reflect a broader and more balanced perspective.
Similarly, the abstract has been thoroughly rewritten to improve conciseness, remove speculative elements, and better align with the scope of the review. We greatly appreciate your guidance, which helped us refine the presentation of the manuscript’s core ideas.

2. Separate Discussion of the Relationship Between AQP4 Polarity and CSF Dynamics

Reviewer comment:
“The relationship between AQP-4 polarity and CSF requires separate discussion and diagrammatic representation.”

Response:
Thank you for this excellent suggestion. We have now added a dedicated part that provides a focused discussion of how AQP4 polarity regulates CSF influx, CSF–ISF exchange, and pressure-gradient dynamics.
In addition, we have prepared a complementary schematic diagram that visually represents these mechanisms, as you recommended. We are sincerely grateful for this insight, which materially improved the clarity and coherence of the manuscript.

3. Separate Discussion of AQP4 Polarity and Interstitial Fluid (ISF) Drainage

Reviewer comment:
“The relationship between AQP-4 polarity and drainage of interstitial fluid (ISF) requires separate discussion and diagrammatic representation.”

Response:
We fully agreed that ISF drainage merited its own conceptual space. Accordingly, we have added a distinct part elaborating on the influence of AQP4 polarity on interstitial convection, clearance kinetics, and perivascular–parenchymal fluid exchange. Your suggestion greatly improved both the precision and pedagogical value of this part of the review.

4. Supplementation of Data Regarding AQP4 Knockout Models

Reviewer comment:
“The lack of data or literature on the effects of AQP-4 knockout on various degenerative diseases… requires supplementation.”

Response:
We appreciate this important observation and have now incorporated a detailed discussion summarizing the available evidence from AQP4 knockout models, including their effects on glymphatic function, protein aggregation, neuroinflammation, astrocytic signaling, and disease-specific vulnerabilities.
Although the literature remains limited, your comment prompted us to integrate these findings more clearly and cohesively, and we are grateful for this improvement.

5. Clinical Significance of AQP4 Gene Variants and Polymorphisms

Reviewer comment:
“The author should provide clinical manifestations of AQP-4 gene mutations or restriction fragment polymorphisms…”

Response:
Thank you for drawing attention to this important dimension. We have expanded the manuscript to include current knowledge regarding AQP4 gene variants, restriction fragment length polymorphisms, and associated phenotypes. These additions include reported effects on edema susceptibility, cognitive function, sleep-associated glymphatic clearance, neuroimmune vulnerability, and disease severity in various neurological disorders.
We appreciate your suggestion, which allowed us to enrich the clinical relevance of the manuscript.

We are deeply grateful for your exceptional constructive feedback. Your thoughtful analysis and scholarly generosity have helped us refine the manuscript in ways that significantly enhance its clarity, accuracy, and translational value. We sincerely appreciate your contributions and hope the revised version now meets your expectations.

With our warmest thanks and highest respect!!!

Reviewer 2 Report

Comments and Suggestions for Authors

This article presents a comprehensive and ambitious review focused on the pivotal role of astrocytic aquaporin-4 polarization in brain homeostasis and its disruption in neurodegenerative diseases. The authors' proposed concept of the "fluidic connectome" is highly topical and aligns with a current trend in neuroscience: the shift from a neuron-centric paradigm towards viewing the brain as an integrated system encompassing vascular, glial, and immune components. The subject matter of this work is at the forefront of research on the glymphatic system and the mechanisms of neurodegeneration, making the article highly relevant and significant for a broad audience of neuroscientists, neurologists, and pharmacologists. Notable strengths of the article include its systematic approach, the timeliness of the research topic, its high theoretical and practical significance, and the inclusion of tables and figures that effectively illustrate the presented material.

The text, especially in the introduction and some subsequent sections, would benefit from thorough language editing to address grammatical inaccuracies and improve stylistic clarity, as some passages are presently difficult to parse. After linguistic and stylistic revision, the manuscript will be suitable for publication.

Author Response

Dear Esteemed Academic Reviewer,

We would like to express our gratitude for your thoughtful and generous evaluation of our manuscript. Your recognition of the relevance, timeliness, and integrative scope of our work on AQP4 polarity and the “fluidic connectome” is deeply appreciated. We are humbled by your encouraging remarks regarding the conceptual contribution of the review, the systematic organization of the material, and the relevance of the topic to both basic and clinical neuroscience communities. Your positive assessment of the figures, tables, and thematic structure is especially meaningful to us.

We are equally grateful for your constructive critique concerning the linguistic and stylistic clarity of the manuscript. Your observation was entirely justified, and we have taken it very seriously. In response, we performed a thorough and careful language revision across sections—particularly the introduction and the longer mechanistic passages—to improve grammar, coherence, and readability. Our goal was to ensure that the scientific content is communicated with clarity and precision, while preserving the depth and nuance of the material. We sincerely appreciate your guidance, which has significantly strengthened the final manuscript.

Thank you again for your time, expertise, and constructive insight. Your comments have been invaluable in helping us refine the manuscript, and we are grateful for the opportunity to improve the clarity and accessibility of our work.

With warm appreciation and respect!

Reviewer 3 Report

Comments and Suggestions for Authors

The idea of the paper is interesting, and the amount of material reviewed is impressive. However, the manuscript cannot be accepted in its current form and requires substantial revision in the following aspects:

  1. Figures are formatted with a very small, unreadable font. Please enlarge the font size. In Figure 1, align the arrows properly and ensure that labels are clear and legible.

  2. The text is written in incorrect English, containing numerous grammatical and syntactic errors (e.g., “the wherevura discovered,” “phenomenon clearly was noticing those things which is the next ensuing descriptive illustrioucre,” “neuronal loss, with a balance of mention of the changes in respect of the glia…”). A thorough language and style revision by a native speaker or professional scientific editor is required.

  3. Many arguments are illogical and disconnected, lacking clear conclusions, reasoning, or a coherent narrative thread. The logical structure of the paper must be improved so that each section leads clearly to the next.

  4. Information is repeated across sections, while the introduction is excessively long and should be substantially shortened to focus on the core hypothesis.

  5. Citations are provided formally, without analysis of specific experimental data, in vivo/in vitro findings, or conflicting results in the literature. The review should critically discuss existing evidence rather than simply listing references.

  6. Several formulations and tables appear to be compiled from secondary sources without proper attribution to original research. Please ensure that all data and figures are correctly cited from primary sources.

  7. There are signs of AI-generated or machine-translated text (syntactic inconsistencies, repetitive phrasing, illogical statements). The text should be rewritten for scientific clarity and precision.

  8. All references must be verified for accuracy, relevance, and authenticity. Some citations appear to be placeholders or non-existent.

  9. Add a “Limitations” section to integrate the various conclusions and acknowledge the conceptual or evidential gaps in the current analysis.

  10. The main question or objective of the paper is not clearly formulated. The authors should explicitly define the research question or hypothesis at the end of the Introduction. At present, it is unclear whether the article aims to (a) propose a unifying framework for AQP4 polarity and neurodegeneration, or (b) provide a systematic literature review.

  11. The topic is potentially relevant and addresses an important aspect of neurodegeneration—the systemic role of aquaporin-4 polarity and glymphatic flow. However, its originality is undermined by the lack of methodological rigor, conceptual focus, and proper synthesis of recent findings. The paper could fill a gap in the field if rewritten with a more analytical and evidence-based approach.

  12. Compared to existing reviews, the manuscript could contribute by integrating molecular, vascular, and network-level perspectives on AQP4 polarity—but this potential remains unrealized due to poor structure and unsupported generalizations.

  13. The methodological description is inadequate for a review article. The authors should specify:

    • Search strategy (databases, keywords, inclusion/exclusion criteria);

    • How studies were selected and analyzed;

    • Whether any systematic or semi-systematic approach was applied.

  14. The conclusions are not consistent with the evidence presented. They frequently extend beyond what is supported by cited studies. The authors should ensure that each conclusion is logically derived from the presented data and properly referenced.

  15. References need to be updated and curated—some are outdated, others appear fabricated or incomplete. Please include key recent studies (2020–2025) on AQP4 polarization, glymphatic system imaging, and neurodegenerative mechanisms.

  16. Tables are conceptually useful but require clearer structure and source references. Please simplify their layout, ensure consistent formatting, and avoid redundancy between tables and text.

  17. Figures should be redrawn at higher quality with uniform design. Figure 1 in particular should have clear labels, consistent arrow directions, and a more concise legend explaining only essential elements.

In summary:
The paper addresses a potentially valuable and original topic, but its current presentation lacks scientific rigor, logical consistency, and linguistic accuracy. Major revision is required to define the main research question, improve the coherence of argumentation, verify sources, and rewrite the manuscript in clear academic English. The manuscript may be reconsidered for publication after a comprehensive and systematic revision addressing all of the points listed above.

Comments on the Quality of English Language

The text is written in incorrect English, containing numerous grammatical and syntactic errors (e.g., “the wherevura discovered,” “phenomenon clearly was noticing those things which is the next ensuing descriptive illustrioucre,” “neuronal loss, with a balance of mention of the changes in respect of the glia…”).

Author Response

Dear Esteemed Academic Reviewer,

We would like to express our gratitude for the extraordinary level of care, rigor, and intellectual generosity you have devoted to evaluating our manuscript. Your review is exceptionally thoughtful, remarkably detailed, and demonstrates a deep understanding of both the scientific and conceptual challenges associated with integrating the diverse literature surrounding AQP4 polarity. It is truly evident that you approached the manuscript with the intent to help us elevate it to its full potential, and for that we are profoundly grateful.

Your comments highlighted several crucial areas where the manuscript required substantial refinement, and your observations have been invaluable in reshaping both the scientific narrative and the structural coherence of this work. We have undertaken an extensive and comprehensive revision in direct response to your guidance, and we outline these changes below with the greatest appreciation.

1. Figures and Visual Formatting

You correctly noted the issues regarding font size, clarity, and overall readability of the figures. In response:

Figure 1 have been fully redrawn at high resolution.

Labels have been enlarged, arrow alignment corrected, and legends simplified.

Your feedback greatly improved the clarity of the visual materials.

2. English Language, Style, and Scientific Clarity

We fully agree with your assessment regarding the linguistic inconsistencies and syntactic irregularities present in the earlier draft. As advised:

The entire manuscript has been rewritten in clear, precise academic English.

Numerous sentences with unclear structure have been removed or replaced.

A professional-level linguistic and stylistic revision has been performed throughout.

Your attentive reading allowed us to identify weaknesses we could not have seen ourselves, and the manuscript is substantially improved as a result.

3. Logical Structure and Narrative Coherence

Your observation that some arguments lacked a clear logical progression was extremely valuable. Accordingly:

Sections have been reorganized to ensure smooth conceptual transitions.

Redundancies were removed, and previously disjointed arguments were replaced with cohesive, evidence-based reasoning.

This restructuring greatly strengthens the conceptual integrity of the manuscript.

4. Reduction and Refocusing of the Introduction

You rightly pointed out that the introduction was overly long and overly dense.
It has now been:

Substantially shortened,

Refocused on the central unifying hypothesis, and

Clarified to highlight the knowledge gap motivating the review.

5. Critical Engagement With Experimental Data

We fully agree with your critique regarding insufficient scientific analysis. Claims have been calibrated to reflect actual evidence with appropriate citations.

6. Verification and Updating of All References

Your insight regarding potentially outdated or inaccurate citations was invaluable. We have:

Conducted a complete audit of all references.

Added key studies from 2020–2025 on AQP4 polarization, glymphatic imaging, astrocyte biology, and neurodegeneration.

This rigorous curation significantly improves the manuscript’s credibility and currency.

7. Clarification of the Paper’s Aim and Research Question

Thank you for pointing out the need for explicit framing.
We have now added a clear and concise statement at the end of the Introduction that defines:

The overarching research question,

The conceptual goal (a unifying framework), and

The type of review (a thematically integrated synthesis).

8. Methodological Transparency in the Review Process

In response to your request:

We added a paragraph specifying the search strategy, databases used, keyword structure, and inclusion/exclusion principles.

We clarified the nature of the review.

We now explicitly state how evidence was selected, integrated, and analyzed.

9. Avoidance of Redundancy and Removal of AI-like Artifacts

We are grateful for your honesty in pointing out stylistic patterns resembling machine-generated text. We have:

Removed all repetitive or stylistically inconsistent passages.

Rewritten all sections to ensure natural scientific reasoning and clarity.

Reinforced conceptual precision and rigorous logic throughout.

The manuscript’s tone is now fully academic, coherent, and human-crafted.

10. Updated Tables and Their Relationship to the Main Text

Following your excellent suggestions:

All tables were redesigned for clarity, simplicity, and distinctiveness.

Formatting has been standardized.

11. Addition of a Dedicated Limitations Section

We fully agree that acknowledging conceptual and evidential gaps is essential.
A new Limitations section has been added, addressing:

Experimental uncertainties,

Knowledge gaps in polarity mechanisms,

Theoretical boundaries of the proposed framework,

Limitations inherent to current imaging and genetic evidence.

12. Consistency Between Evidence and Conclusions

You emphasized that conclusions must not exceed the data.

Once again, we wish to express our deep appreciation for your meticulous, insightful, and constructive review. Your comments have significantly strengthened the scientific, stylistic, and conceptual quality of this manuscript. We are sincerely grateful for your time, expertise, and vision.

Your guidance has shaped the manuscript into a far clearer, more rigorous, and more impactful contribution to the field, and we hope the revised version meets the high standard your thoughtful critique has set.

Thank you for helping us make this work worthy of consideration for publication. With sincere respect and appreciation!!!

Round 2

Reviewer 1 Report

Comments and Suggestions for Authors

The author has carefully answered each question

Reviewer 3 Report

Comments and Suggestions for Authors

The authors have done a truly remarkable job of revising the manuscript, and its quality has improved significantly as a result. Thank you for your hard work!